# Does Participation in Sports Influence the Prevalence of and Initiation into Multiple Substance Misuse in Adolescence? A Two-Year Prospective Analysis

**DOI:** 10.3390/children7090109

**Published:** 2020-08-22

**Authors:** Natasa Zenic, Martina Rezic, Ivana Cerkez Zovko, Hrvoje Vlahovic, Tine Sattler

**Affiliations:** 1Faculty of Kinesiology, University of Split, 21000 Split, Croatia; 2Faculty of Science and Education, University of Mostar, 88000 Mostar, Bosnia and Herzegovina; martina.rezic@fpmoz.sum.ba (M.R.); ivana.cerkez.zovko@fpmoz.sum.ba (I.C.Z.); 3Faculty of Health Studies, University of Rijeka, 51000 Rijeka, Croatia; hrvoje.vlahovic@uniri.hr; 4Faculty of Sport, University of Ljubljana, 1000 Ljubljana, Slovenia; tine.sattler@fsp.uni-lj.si

**Keywords:** physical exercise, puberty, substance misuse, risk factors, protective factors, sport

## Abstract

Concurrent smoking and harmful drinking (CSHD) in adolescence is an important public health and social problem, while participation in sports is considered as being protective against CSHD. This study aimed to prospectively evaluate the influence of various facets of sports participation on the prevalence of and initiation into CSHD of adolescents. Participants were adolescents from southern Croatia (*n* = 711, 43.6% females, 16 years of age at study baseline), who were tested at baseline and at follow-up (two years later). Variables included gender, age, sports factors (participation in individual and team sports, sport experience, competitive success, intensity of involvement in sports), and CSHD. The CSHD prevalence did not increase significantly over the course of the study (from 5.6% to 7.5%, *p* > 0.05). Binomial logistic regression with age and gender as covariates suggested that team sports participation correlated to CSHD prevalence at baseline, and follow-up, with higher risk for CSHD among those adolescents who quit team sports (OR = 9.18 and 2.68, 95%CI = 2.04–22.26 and 1.05–6.83 for baseline and follow-up, respectively), and those never involved in team sports (OR = 9.00 and 3.70, 95%CI = 2.07–39.16 and 1.57–8.72 for baseline and follow-up, respectively). A higher risk of CSHD at baseline was seen among those adolescents who were involved in sports for longer (OR = 1.66, 95%CI = 1.16–2.38). The results are discussed in the context of the fact that the study included adolescents at the age of rigid sports selection (the transition from youth to professional-level sports). Since the majority of participants began CSHD at an earlier age, further studies in subjects of a younger age range are warranted.

## 1. Introduction

Although it has been decreasing globally, cigarette smoking remains one of the leading preventable causes of death worldwide [1,2]. The problem is aggravated in vulnerable populations and countries where there is no social stigma against smoking [3,4]. This is particularly important for adolescents, because early initiation into smoking is a risk factor for later nicotine dependence. With more than 30% of adolescents smokers, countries of southeastern Europe, including Croatia, are among the European regions with the highest prevalence of adolescent smoking [5]. Such figures have led to intensive scientific effort and professional debate aimed at the reduction in smoking in adolescents from southeastern Europe (the former Yugoslavia) over the last 10 years [6,7,8,9]. Indeed, recent figures showed promising trends in terms of a decrease in smoking prevalence among adolescents from the region [10].

Alcohol is also a popular psychoactive substance in southeastern Europe; figures from the European School Survey Project on Alcohol and Other Drugs (ESPAD) regularly show high alcohol consumption among adolescents [5,11]. Culturally, this is explained by various factors, including the Mediterranean lifestyle. Briefly, although not all countries in the region are located on the Mediterranean Sea, the strong influence of Italian and Greek culture and the similarity of the climate (which is, in most countries, regularly sunny and mild) have resulted in the popularity of vine-growing in the whole territory. Therefore, wine consumption is frequent, and adolescents are initiated into wine consumption very early, in a family setting. This, at least partially, results in high numbers of adolescents becoming alcohol consumers [7,12]. Excessive alcohol consumption is a known public health problem, but also results in other serious problems that are not exclusively related to public health (i.e., drunk driving, involvement in fights, violence, and unsafe sex) [13,14]. As a result, the distinction between “harmful” and “non-harmful alcohol drinking” is even recognized by the World Health Organization (WHO), emphasizing the former as being a “problematic” form of alcohol consumption that should undoubtedly be prevented [6,15,16].

While both smoking and harmful drinking (HD) are known to be problematic and dangerous habits, concurrent smoking and harmful drinking (CSHD) is especially hazardous [13,17]. CSHD exacerbates the consequences of smoking and HD. For example, CSHD is particularly dangerous when operating motor vehicles (i.e., alcohol is a depressant, and smoking negatively influences one’s ability to react properly). In addition, while one drug is a depressant and the other one is a stimulant, people who simultaneously drink and smoke may not realize how much the alcohol is affecting their body, which could cause them to drink more because they do not feel drunk. Recent studies showed that drinking and smoking together can increase the risk of throat and esophageal cancer (i.e., alcohol dissolves chemicals in the cigarette while they are still in the throat). Last but not least, CSHD in adolescence has been identified as a risk factor for later alcohol dependence [13,18].

It is well known that participation in sports and physical exercise provides numerous benefits to youth, including a reduced risk in the development of obesity, improved motor competence, better cardiovascular fitness, and superior conditioning capacities [19,20]. It has also been suggested that participation in sports and exercise may prevent youths from various problematic behaviors, including the consumption of psychoactive substances such as alcohol and cigarettes. The hypothesis of the protective effects of participation in sports during adolescence against smoking and drinking is based on the fact that sports promote overall well-being, and positively influence the development of various prosocial behaviors and self-discipline [21]. Furthermore, sports participation has certain characteristics that should have a direct protective effect against smoking/drinking initiation (i.e., orientation toward success, age-bonding, parental supervision) [22]. Logically, it is expected that adolescents who are involved in sports are less likely to smoke and drink [23]. However, the empirical evidence on this topic is not consistent.

The prevalence of smoking is regularly lower in adolescents who practice sports, but some groups of athletes (e.g., adolescents involved in sports where a lean body figure is an important determinant of success, females, and adolescents who quit sports) were found to be at risk of smoking [24,25]. In addition, in some regions where smoking is prevalent and socially accepted, no specific influence of sports participation on smoking prevalence in adolescents was observed [8,26,27]. When it comes to alcohol and sports participation, the situation is even more complicated. Some authors reported a lower likelihood of alcohol consumption in adolescents involved in organized sports [14], but for each study where protective effects of sports against alcohol were found, another one evidenced an increased risk of alcohol consumption in athletic adolescents [6,28,29,30]. According to recent findings, it seems that sports participation may not be simplified when observed as a factor of influence on drinking, so various facets of sports participation should be evaluated independently in order to more precisely identify the complex relationships that may exist between sports and the consumption of various psychoactive substances in adolescence [9,10]. This is particularly evident with regard to participation in individual (i.e., athletics, swimming) and team sports (i.e., football/soccer, basketball, handball), and studies repeatedly found differential effects of participation in these two type of sports on substance misuse in adolescence [9,31]. In brief, participation in team sports was repeatedly associated with higher likelihood of alcohol drinking (because of the post-sport social gatherings) [32]. On the other hand, athletes involved in some individual sports were often at risk of smoking (i.e., in some aesthetic sports athletes are concerned on body weight, because smoking increases basal metabolism) [24,25]. Other facets of sports participation that are found as being specifically related to substance misuse in puberty include experience, level of sport competition (i.e., sport achievement), and intensity of training (i.e., commitment to sport), altogether highlighting the necessity for more profound studying of the problem [7,27,31].

Collectively, despite the overall importance of sports participation and the prevention of substance misuse in adolescence, we are still witnessing a gap in the literature when it comes to explaining the complex relationships that exist between them. The problem of CSHD is especially understudied, and the majority of studies that examined sport-participation in adolescence as factor of influence on CSHD were cross-sectional [33]. In addition, there is a lack of research that prospectively evaluates the relationship between sports participation and CSHD in adolescence. Finally, to the best of our knowledge, no study has so far examined this problem in southeastern Europe, which would be particularly important knowing the alarmingly high rates of substance misuse among adolescents from this region [6,9,31]. Therefore, the aim of this prospective study was to evaluate the causal effects that exist between various factors explaining sports participation and CSHD in adolescence. Initially, we hypothesized that the studied sports factors would be positively associated with CSHD prevalence and CSHD initiation in the studied adolescents.

## 2. Materials and Methods

### 2.1. Participants

The participants in this study were adolescents from three regions/counties in Croatia; Split-Dalmatia, Zadar, and Sibenik-Knin (see Figure 1). The sampling was based on a multistage cluster sampling method, including (i) random selection of one-fourth of high schools in the three studied counties and (ii) random selection of one-third of the third-year classes in the selected schools. The sizes of the schools varied up to 15% and therefore schools were not stratified by size. Of the 821 eligible participants, 711 had submitted all the necessary data in both the baseline and follow-up testing and were included in this study. Approval from the ethical board of the Faculty of Kinesiology, University of Split, Croatia (EBO: 2181-205-05-02-05-14-005), was obtained prior to the start of the study. Additionally, the investigation was authorized by the school authorities.

### 2.2. Procedures

Participants were observed on two occasions: at the study baseline (first wave, when they were in their third year of high school (16 years old on average); September 2016) and again at the follow-up (approximately 18 months later, at the end of the fourth year of high school; May 2018). Before testing, all participants were informed about the study and the investigators provided a written explanation of the study’s background, aims, and protocol to the children’s parents, who gave written consent for their children’s participation. At both baseline and follow-up, the participants were informed that they could leave questions unanswered. No personal information was requested, and the participants remained anonymous. However, the participants used confidential codes to assure tracking in the repeated test. The testing was performed through an internet-based application. Investigators prepared the online survey in advance and tested participants during the regular school hours. Participants responded to the survey using their own cell phones. During the testing, investigators had a couple of mobile phones that were used for responding to the survey if it was necessary.

The analysis of attrition bias showed no significant differences in the initial CSHD status between those participants who remained in the study and those who dropped out (χ^2^ = 0.98, *p* > 0.05). However, significantly more females remained in the study protocol than males (χ^2^ = 9.98.00, *p* < 0.01). The intracluster correlation coefficient calculated for prevalence of CSHD at baseline with schools observed as clusters was 0.076, indicating appropriate within-school variance [34].

### 2.3. Instruments and Variables

In both testing stages, participants were tested using the previously validated questionnaires: the Alcohol Use Disorders Identification Test (AUDIT) and the Questionnaire of Substance Use (QSU) [12,35,36].

The QSU is a questionnaire already found to be a reliable and valid measuring tool in southeastern Europe (i.e., the former Yugoslavia) [12,26,27]. Among other factors, QSU evaluates sociodemographic and sports factors, and data on substance misuse. The sociodemographic factors were participants’ gender (responses included: male—female—other) and age (in years). The sports factors included in this study were assessed via four questions examining different facets of sports involvement, namely: (i) involvement in team sports, (ii) involvement in individual sports (both responded on a scale including: never been involved, quit, currently involved); (iii) highest competitive achievement in sports (responses included: never competed/did not participate in sports, local competitions, national/international-level competitions); (iv) time of involvement in sports (including: never involved, <1 year, 2–5 years, >5 years); (v) number of training sessions per week (for those involved in sports; responses included: 1–2, 3–5, >5).

Questions on substance misuse included queries aimed at the evaluation of cigarette smoking and drinking. Smoking was queried with one item (never smoked, quit, smoking but not daily, daily smoking), and responses were later dichotomized into two clusters (nonsmokers (first two responses) vs. smokers). The AUDIT questionnaire measured the consumption of alcohol. This questionnaire contains 10 items with scores ranging from 0 to 4 for a hypothetical minimum (0) to maximum (40). For the purposes of this study, we separately observed three domains of the AUDIT [36]. The overall results of the AUDIT were divided into “harmful drinking” (HD) and “non-harmful drinking” (NHD), and this classification was used for the logistic regression calculation. For a meaningful comparison with the results from previous studies on similar subject samples, we used a total score of 11 as a cutoff score for HD, although two types of HD and NHD dividing scores are suggested in the literature (i.e., using the scores of 8 and 11 as cutoff points) [6,12,37]. Those participants who reported smoking and HD were categorized into a group of concurrent smokers/harmful drinkers (for later statistical analysis numbered “2”, and “1” otherwise).

### 2.4. Statistics

The normality of the distributions was checked by the Kolmogorov‒Smirnov test. For all the variables, descriptive statistic counts and percentages were calculated. Depending on the characteristics of the variable, the differences between the groups (Males vs. Females; CSHD vs. non-CSHD) were established by a Mann‒Whitney test (for ordinal variables), or the Chi-square test (χ^2^; for categorical variables). Binary logistic regression was used to estimate the odds ratio (OR) and the corresponding 95% confidence interval (95%CI) of the following: (i) CSHD at the baseline, (ii) CSHD at the end of the study, and (iii) CSHD initiation occurring during the course of the study by the studied covariates (for this calculation, we included only those adolescents who were non-CSHD at baseline). The logistic regression calculations were adjusted for gender and age because previous studies identified significant influence of age and gender on substance misuse in adolescence [6,9] The model fit was checked by the Hosmer Lemeshow test (with significant χ^2^ indicating inappropriate model fit). Statistica ver. 13.5 (Tibco Inc., Palo Alto, CA, USA) was used for all calculations, with the significance level of *p* < 0.05 applied for all calculations.

## 3. Results

Data on prevalence of smoking and harmful alcohol drinking are presented in Appendix A. The prevalence of CSHD at baseline and follow-up is presented in Figure 2. In the total sample, the prevalence of CSHD increased from 5.6% at baseline, to 7.5% at follow-up (χ^2^ = 1.68, *p* = 0.15). The prevalence of CSHD was higher in boys than in girls at both testing waves (χ^2^ = 13.75 and 19.21, *p* < 0.001, for baseline and follow-up, respectively).

Descriptive statistics of the studied sport factors, and differences between groups according to CSHD prevalence are presented in Table 1. The prevalence of CSHD was higher in adolescents who practiced individual sports, and team sports, both at baseline and follow-up. Similarly, sport experience, sport achievement and training frequency were correlated with CSHD prevalence. However, all stated was certainly influenced by the fact that prevalence of CSHD was higher in boys than girls (Figure 1), while sport participation was higher in boys (Appendix A).

Correlates of CSHD at baseline, follow-up and CSHD initiation are summarized in Table 2. Higher likelihood for CSHD at baseline and follow-up was evidenced in adolescents who quit team sports (OR = 9.18 and 2.68, 95%CI = 2.04–22.26 and 1.05–6.83 for baseline and follow-up, respectively). Additionally, those adolescents who never participated in team sports were at greater risk of CSHD at baseline and at follow-up than those who were currently involved in team sport (OR = 9.00 and 3.70, 95%CI = 2.07–39.16 and 1.57–8.72 for baseline and follow-up, respectively). Longer experience in sports was found as a risk factor for CSHD at baseline (OR = 1.66, 95%CI = 1.16–2.38). Studied sport factors were not correlated to CSHD initiation.

## 4. Discussion

This study aimed to evaluate the prevalence of CSHD, and sport factors associated with CSHD in adolescents from Croatia. Our results indicated several important findings. First, the prevalence of CSHD did not change significantly over the study period (two years), and in both testing waves the prevalence was higher among boys. Next, sports participation was specifically related to CSHD prevalence with: (i) current sports participation being a protective factor against CSHD, and (ii) longer sports participation being a risk factor for CSHD at both testing points. Finally, the studied sports factors were not significantly associated with CSHD initiation among the studied adolescents. Therefore, we can only partially accept our initial study hypothesis.

### 4.1. Prevalence and Dynamics of Changes in CSHD in Adolescence

Although we could not find any study performed in southeastern Europe where trends in changes in CSHD were reported, we may say that the increase in the prevalence of CSHD prevalence that we found for those between 16 and 18 years of age in this period was expected. Namely, recent studies in the former Yugoslavia reported a significant increase in the smoking prevalence in Bosnian‒Herzegovinian and Croatian adolescents over a two-year period (from 40% to 47% ever smokers) and from 28% to 36% (daily smokers) for Croatian and Bosnian‒Herzegovinian adolescents, respectively) [10,31]. In addition, a significant increase in drinking prevalence was recorded by the AUDIT score in adolescents from Bosnia and Herzegovina for the same age [9]. Altogether similar increases are evidenced in our study as well. Although previous studies did not report the prevalence of CSHD, it is logical to expect that the increase in prevalence of smoking and drinking will increase CSHD between the ages of 16 and 18 years. While the possible reasons for such dynamics are complex, and an explanation goes beyond the scope of this paper, more attention should be paid to sports as a possible factor of influence on such changes.

The CSHD prevalence was higher in boys, but girls experienced a relatively higher increase in CSHD prevalence in the studied period (boys: from 8.6% to 10.4%; girls: from 2.0% to 3.6%). Therefore, the prevalence of CSHD among boys increased by 25%, while the prevalence of CSHD in girls practically doubled in the studied period. These results can be explained in light of previous reports. Briefly, Zenic et al. reported a two-fold increase in harmful drinking in both adolescent boys and girls over a two-year period [9]. On the other hand, another study reported a much lower increase in smoking prevalence over two years in adolescent boys than in girls (girls: 37% to 49%; boys 44% to 48%) [31]. Our results, however, emphasized another interesting conclusion. In brief, the increase in CSHD can be logically expected as a direct consequence of (i) increased smoking, and (ii) increased harmful drinking. However, based on our results, it seems that changes in smoking prevalence are more influential on CSHD in the studied period of life, at least for older adolescents. In fact, this directly supports previous reports wherein smoking initiation and persistence were found to be a function of prior drinking (while vice versa was not so pronounced) [13].

### 4.2. Sports Factors and Multiple Substance Misuse in Adolescence

Before discussing the influence of sports on CSHD in the studied adolescents, we note that gender was regularly found to be an important (although not always significant) covariate of established relationships. For a better understanding of those findings, it is important to note that boys are more involved in sports and had a higher prevalence of CSHD than girls. Both statements are supported by previous studies, in which authors regularly reported (i) higher sports participation and (ii) more frequent substance misuse in boys than in girls of the same age from the region [6,8,31]. As a result, such “trends” are logically translated into relationships between sports factors and CSHD in our study as well, highlighting gender as an important covariate in established correlations. On the other hand, sports factors are still found to be a strong factor of influence on CSHD in our participants, while the specifics of those associations indicate the appropriateness of our methodological approach where we tried to overview different facets of sports participation as possible factors of influence on CSHD.

The fact that in both testing periods active participation in team sports was found to be protective against CSHD is one of the most important findings of our study. More specifically, although gender was a significant confounding factor, even when the logistic regression calculation included gender as a covariate, a higher likelihood of CSHD was found in those adolescents who never participated in team sports, and those who had participated but quit. For a better understanding of the results, we will shortly survey the findings of those reports in which smoking and drinking were prospectively observed as “consequences” of sports participation in adolescents [7,9,31].

Specifically, in a study with Croatian adolescents, Devcic et al. showed that those adolescents who were currently involved in, or had participated in but later quit, team sports were almost twice as likely to be engaged in harmful drinking as their peers who had never practiced team sports [7]. Similar findings were reported in a study that prospectively examined adolescents from Bosnia and Herzegovina [9]. However, with regard to smoking, only those adolescents who had quit team sports were at higher risk of smoking than those who never practiced team sports [31]. Putting it all altogether, previous reports could lead to the assumption that team sports are more likely to be a “risk factor” than a “protective factor” for substance misuse, and logically this could be applied to CSHD as well. However, it seems that active team sports athletes are not likely to engage in CSHD.

For a more detailed elaboration of the problem, the findings of a comprehensive study of adult athletes involved in a specific team sport (rugby) are interesting [32]. In brief, in that study the authors examined all adult male athletes from Croatia involved in rugby and discovered a very specific behavioral template consisting of (i) regular alcohol consumption (which was actually alarming) and (ii) avoidance of cigarettes. The explanation for such specific figures was based on a specific “culture” that globally exists in rugby whereby alcohol consumption is a frequent and actually unavoidable part of life for this social group [38,39]. Meanwhile, rugby athletes avoid cigarettes since they are aware of the negative effects of smoking on their physical performance. Despite some differences between studies (i.e., adolescents vs. adult athletes, team sports. vs. rugby), the “logic” provided in the rugby study is probably transferable to our sample as well. In brief, although the team sports culture often goes along with the social acceptability of drinking, this does not necessarily mean that team sports athletes who consume alcohol will concurrently smoke cigarettes.

Interestingly, participation in individual sports was not found to be related to CSHD. This is intriguing since previous studies regularly showed a stronger protective effect of participation in individual sports than of team sports against substance misuse in adolescence [6]. This was explained by the stronger “orientation to success” and “commitment” in children and adolescents who participate in individual sports. The lack of association in our study may be explained from at least two perspectives. First, it is possible that the relatively small number of adolescents who were engaged in CSHD, together with the small number of adolescents who had certain experiences in individual sports, resulted in the small number of subjects in the corresponding groups, and statistically resulted in the insignificant association between individual sports participation and CSHD [40]. A second explanation is more “sport-specific” and asks for a more profound elaboration of the characteristics of the individual sports.

Individual sports have very different characteristics. While some individual sports require a lean body figure (i.e., aesthetic sports such as dance and artistic gymnastics), in others the required body build is totally the opposite (i.e., martial arts in higher weight categories, some technical disciplines in athletics) [41]. Furthermore, in some individual sports athletes are highly concerned about their cardiovascular endurance (i.e., “running disciplines” in track and field athletics, swimming), while in some others cardiovascular endurance is not among the top fitness priorities (weightlifting, martial arts, dance). This complicates the tendencies towards the consumption of psychoactive substances. More precisely, studies have already shown that athletes in aesthetic sports (mostly females) frequently smoke cigarettes in order to increase their basal metabolism, and/or because they are convinced that smoking acts as an appetite suppressant [25]. On the other hand, alcohol consumption in these sports’ participants is not frequent simply because of the caloric value of the alcohol [25]. Meanwhile, athletes involved in martial arts do not avoid alcohol, a substance already recognized as being a part of their “sociocultural milieu”, similar to what was previously discussed for rugby [32]. Together with the previously specified “statistical issue” (i.e., the relatively small number of participants in corresponding groups (individual sports participants vs. CSHD consumers)), this probably explains why we did not find evidence of any relationship between these variables.

One of the most disturbing results of the study is the higher risk of CSHD seen for those adolescents with longer experience of sports. Although it may seem surprising, this finding is actually in agreement with previous studies where sports factors were correlated with drinking in adolescents [7,8]. Meanwhile, to the best of our knowledge, this is one of the first studies where sports experience was directly found to be a risk factor for multiple substance misuse in adolescents. This finding demands a profound investigation of the characteristics of sports participation in the studied age group.

The ages of 16 to 18 represent the most stressful period in young athletes’ lives since this is when the systematic selection between “youth sports” and “professional sports” occurs. In other words, only those adolescent athletes who are recognized as being predisposed to top-level achievement are retained for systematic training and competitions. It is not hard to recognize the stress that is placed on youths who are involved in sports and not encouraged about their capacities, abilities and qualities, which would hopefully allow them to continue their sporting career. It is also logical that such stress is even more pronounced among those adolescents who have been engaged in the sport for a long(er) time, simply because the sport becomes an essential part of their life. Most probably, one of the consequences of such stress is the higher likelihood of CSHD. Collectively, this is supported by studies in which a higher likelihood of doping behavior is seen in athletes of that age [42].

One could argue that at least those adolescents who achieved “sporting success” should be protected against CSHD. Although this is a logical assumption, this phenomenon is unlikely to occur in real life for two reasons. First, even those adolescents who are successful in sports are stressed in this age [43], and therefore may lean toward CSHD as much as their unsuccessful peers. Second, even if we accept the possibility that “high achievement” may be protective against CSHD, the problem is the relatively low percentage of high achievers in a population of adolescents who were involved in sports for a relatively long period. Specifically, of all the studied adolescents who were involved in sports for more than five years (25%), only a minority (<3%) achieved highly competitive results, and such a small number of the “highly successful” would not likely reduce the previously discussed correlation between sports experience and CSHD.

None of the studied sports factors were significantly correlated with CSHD initiation in the studied period, meaning that sports should not be considered a risk or a protective factor in terms of CSHD initiation between the ages of 16 and 18. Most probably, the fact that the majority of adolescents who reported CSHD at the follow-up (when they were 18 years old) began CSHD earlier in life (i.e., before the baseline measurement) limited the possibility of finding significant relationships simply because of the disproportionate number of subjects in the dichotomized criterion (CSHD vs. non-CSHD). Additionally, it is possible that some factors other than those studied here influenced CSHD initiation to a greater extent. Regardless, such correlations should be studied earlier in life when adolescents initiate smoking and drinking (i.e., between the age of 14 and 16 years).

### 4.3. Limitations and Strengths

The most important limitation of the study is that substance misuse was self-reported and therefore participants may lean toward socially acceptable responses. However, we believe that the study design, and strict anonymity of the testing, reduced this possibility. In addition, this study was focused solely on sports factors as possible correlates of CSHD, while some important covariates were not included in the study design. Finally, we observed a region where smoking and drinking are socially acceptable behaviors, even in adolescence, so people start to smoke and drink early. Therefore, the results are generalizable, but only to similar populations.

This is one of the rare studies in which sports factors have been systematically correlated with CSHD, and probably the first to examine the problem prospectively. Therefore, the causality of the association may be interpreted clearly. Importantly, we used previously validated measurement tools, which allowed us to accurately compare the results with those previously reported.

## 5. Conclusions

In our study, the prevalence of CSHD did not change significantly between 16 and 18 years of age. However, since the majority of the studied participants started CSHD before age 16, further studies are needed that consider the earlier years.

Participation in team sports was shown to be a protective factor against CSHD both at baseline (16 years of age), and follow-up (18 years of age). Evidently, young athletes involved in team sports avoid simultaneous consumption of various psychoactive substances. From a public health perspective, these results should be disseminated in systematic educational and preventive campaigns against CSHD in adolescence.

Experience of sports was found to be a factor that increased the risk of CSHD. It is likely that those adolescents who were more experienced in sports at this particular age faced huge psychological stress because of the forthcoming sports selection process (the transition from amateur to professional level). This results in an even higher likelihood of multiple substance misuse. There is no doubt that the problem is aggravated by the fact that in the studied country there is no opportunity for organized participation in competitive sports at a recreational level (i.e., amateur competitions).

One important suggestion to sports and public health authorities would be to put special effort into the organization of recreational sports competitions. The authors are of the opinion that such recreational-level competitions would allow adolescents to continue with sports participation even if they are not predisposed continuing at a professional level. Apart from being directly associated with a reduced risk of CSHD in adolescent athletes, the opportunity to practice recreational competitive sports in late adolescence will almost certainly have other health-related benefits.

## Figures and Tables

**Figure 1 children-07-00109-f001:**
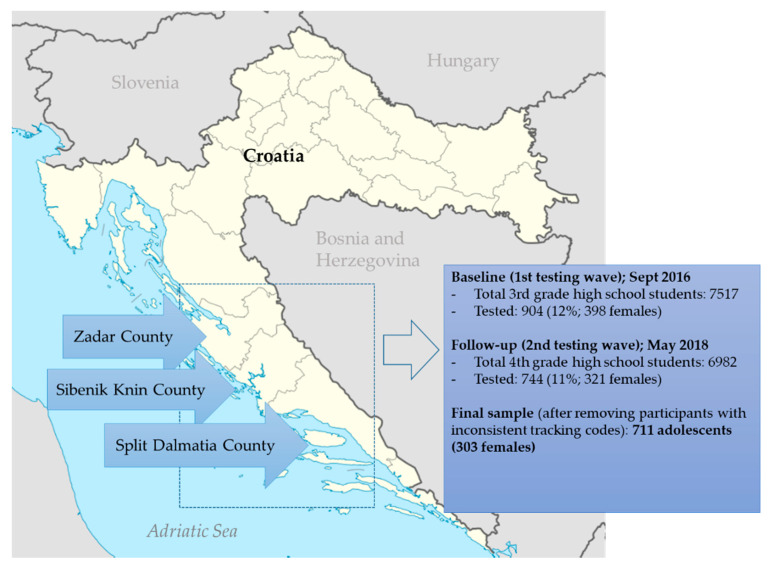
Location of the study, and testing sequences with characteristics of the study sample.

**Figure 2 children-07-00109-f002:**
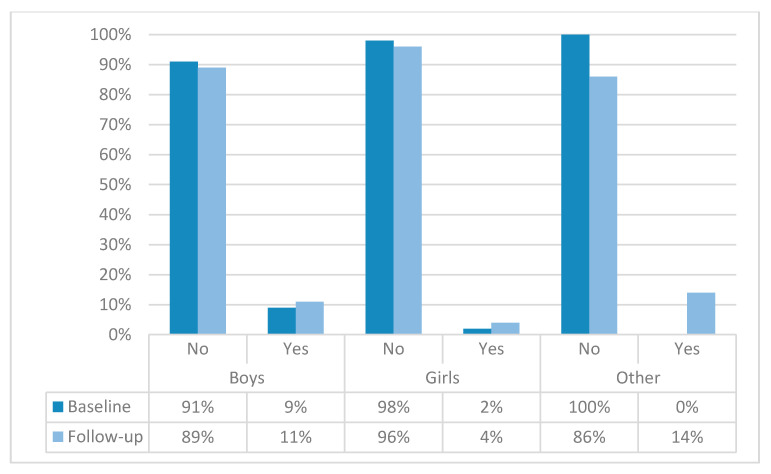
Prevalence of concurrent smoking and harmful drinking in studied adolescents from southern Croatia at baseline and follow-up measurement according to gender.

**Table 1 children-07-00109-t001:** Descriptive statistics for sport factors with differences between groups according to concurrent smoking and harmful alcohol drinking (CSHD) calculated by Mann–Whitney (MW) and Chi square test (χ^2^).

	Baseline Testing			Follow-Up Testing		
	Non-CSHD	CSHD	MW/χ^2^ (*p*)	Non-CSHD	CSHD	MW/χ^2^ (*p*)
	F (%)	F (%)		F (%)	F (%)	
Individual sport participation (χ^2^)			10.79 (0.01)			9.99 (0.01)
Yes, currently	138 (21)	14 (35.9)		134 (20.8)	18 (34.6)	
Yes, but quit	200 (30.5)	16 (41)		196 (30.5)	20 (38.5)	
No, never	318 (48.5)	9 (23.1)		313 (48.7)	14 (26.9)	
Team sport participation (χ^2^)			18.85 (0.001)			14.10 (0.001)
Yes, currently	157 (23.9)	16 (41)		157 (24.4)	16 (30.8)	
Yes, but quit	242 (36.9)	21 (53.8)		234 (36.4)	29 (55.8)	
No, never	257 (39.2)	2 (5.1)		252 (39.2)	7 (13.5)	
Sport experience (MW)			3.38 (0.001)			2.97 (0.001)
Never been involved	193 (29.4)	1 (2.6)		191 (29.7)	3 (5.8)	
<1 year	121 (18.4)	8 (20.5)		119 (18.5)	10 (19.2)	
2–5 years	182 (27.7)	11 (28.2)		170 (26.4)	23 (44.2)	
>5 years	160 (24.4)	19 (48.7)		163 (25.3)	16 (30.8)	
Sport achievement (MW)			2.59 (0.01)			2.74 (0.01)
Never involved/never competed	364 (55.5)	10 (25.6)		358 (55.7)	16 (30.8)	
Local competitions	235 (35.8)	28 (71.8)		231 (35.9)	32 (61.5)	
National/international competitions	57 (8.7)	1 (2.6)		54 (8.4)	4 (7.7)	
Training frequency per week (MW)			3.30 (0.001)			2.43 (0.02)
Not participated in sports	155 (23.6)	1 (2.6)		153 (23.8)	3 (5.8)	
1–2 training sessions	206 (31.4)	10 (25.6)		197 (30.6)	19 (36.5)	
3–5 training sessions	228 (34.8)	22 (56.4)		227 (35.3)	23 (44.2)	
>5 training sessions	67 (10.2)	6 (15.4)		66 (10.2)	7 (13.4)	

**Table 2 children-07-00109-t002:** Correlates of concurrent smoking and harmful drinking at baseline (CSHD-BL), follow-up (CSHD-FU) and initiation of concurrent smoking and harmful drinking during the study period (CSHD-INIT).

	CSHD-BL	CSHD-FU	CSHD-INIT
Individual sport participation			
Yes, currently involved	REF	REF	REF
Yes, but quit	2.27 (0.92–5.55)	2.06 (0.96–4.45)	0.79 (0.27–2.29)
No, never	2.40 (0.93–5.61)	1.96 (0.96–4.03)	1.28 (0.53–3.07)
Team sport participation			
Yes, currently involved	REF	REF	REF
Yes, but quit	9.18 (2.04–22.26)	2.68 (1.05–6.83)	0.95 (0.30–2.98)
No, never	9.00 (2.07–39.16)	3.70 (1.57–8.72)	1.81 (0.71–4.59)
Sport experience ^CONT^	1.66 (1.16–2.38)	1.33 (0.01–1.78)	1.02 (0.72–1.47)
Sport achievement ^CONT^	1.38 (0.88–2.16)	1.49 (0.91–2.20)	1.58 (0.94–2.64)
Training frequency per week ^CONT^	1.44 (0.99–2.08)	1.21 (0.88–1.66)	1.04 (0.67–1.57)

Legend: Results are derived from the logistic regression calculation controlled for gender and age as covariates, CONT denotes variables observed as continuous for the purpose of the logistic regression calculation, data presented are odds ratio (95% confidence interval), REF—reference value.

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
