# Peer review of "Does Participation in Sports Influence the Prevalence of and Initiation into Multiple Substance Misuse in Adolescence? A Two-Year Prospective Analysis"

_children, 2020, doi:10.3390/children7090109_

Round 1

Reviewer 1 Report

The article is interesting due that analyzes the influence of sport participation on the prevalence and initiation in CSHD during the adolescence’ stage. Hence, it provides information to policy managers to create policies to sort out this problem. However, some minor changes should be made in the manuscript before its publication:

Introduction section

In line 94-96, the sentence needs to be clarified because it is difficult to understand the second part of it. It seems as if something was missing. Although the following sentence refers to it, these two sentences are not well connected.

Method section

In line 118, the full stop is missing.

Line 128 indicates that "The testing was performed through an Internet-based application".  It would be necessary to briefly explain what that application was like for passing the test (through the mobile phone, through the computer, in class or at home, etc.).

In line 147, the full stop is missing.

Results section

In figure 2, line 180-181, it is not clear what is meant by "Other", as it is not explained anywhere. Please, explain what “other” means.

In line 183, the full stop is missing.

In Table 2, line 211, the data in the table for "Yes, he is currently involved" is missing for both individual and team sports.

In line 216, the full stop is missing.

In line 235, "CDHD" is written, and should be changed for CSHD.

In line 357, it is stated: "Regardless, such correlations should be studied earlier in life." From what age would it be recommended to study it according to these results? Are there any other studies that justify it?

Reference section

At reference 11 (line 434), the doi is missing.

At reference 14 (line 442), the doi is missing.

At reference 25 (line 472), the doi is missing.

At reference 30 (line 488), the year 2003 has been entered twice.

At reference 33 (line 498), the doi is missing.

At reference 35 (line 503), the doi is missing.

At reference 40 (line 515), revise the citation of this reference.

I recommend the following website to find the articles' doi: https://search.crossref.org/references 

Author Response

REVIEWER 1

The article is interesting due that analyzes the influence of sport participation on the prevalence and initiation in CSHD during the adolescence’ stage. Hence, it provides information to policy managers to create policies to sort out this problem. However, some minor changes should be made in the manuscript before its publication:

RESPONSE: Thank you for recognizing the quality of our work. Also, thank you for accurate and constructive suggestions. We tried to follow it and amended the manuscript. Staying at your disposal.

Introduction section

In line 94-96, the sentence needs to be clarified because it is difficult to understand the second part of it. It seems as if something was missing. Although the following sentence refers to it, these two sentences are not well connected.

RESPONSE: Thank you for noticing it. Inded, text was not fluent. It is changed and now reads:” The problem of CSHD is especially understudied, and majority of studies that examined sport-participation in adolescence as factor of influence on CSHD were cross-sectional“ (please see highlighted text – last paragraph of the Introduction) 

Method section

In line 118, the full stop is missing.

RESPONSE_ Added, thank you.

Line 128 indicates that "The testing was performed through an Internet-based application".  It would be necessary to briefly explain what that application was like for passing the test (through the mobile phone, through the computer, in class or at home, etc.).

RESPONSE: Thank you. Details are added and text reads: “The testing was performed through an Internet-based application. Investigators prepared the online survey in advance and tested participants during the regular school class. Participants responded on survey by their personal mobile phones. During the testing investigators had 5 mobile phones that were used if needed“ (please see 1st paragraoh of the subheading Procedures)

In line 147, the full stop is missing.

RESPONSE: Added, thank you.

Results section

In figure 2, line 180-181, it is not clear what is meant by "Other", as it is not explained anywhere. Please, explain what “other” means.

RESPONSE: The explanation is provided in the Methods section. Text reads: “The sociodemographic factors were participants’ gender (responses included: male – female - other) and age (in years).“ (please see Procedures subsection – 4th paragraph).

In line 183, the full stop is missing.

RESPONSE: Corrected

In Table 2, line 211, the data in the table for "Yes, he is currently involved" is missing for both individual and team sports.

RESPONSE: Indeed, due to formatting of the table data were not evident. It is now corrected. Thank you.

In line 216, the full stop is missing.

RESPONSE: Corrected.

In line 235, "CDHD" is written, and should be changed for CSHD.

RESPONSE: Corrected.

In line 357, it is stated: "Regardless, such correlations should be studied earlier in life." From what age would it be recommended to study it according to these results? Are there any other studies that justify it?

RESPONSE. Details are added and text now reads: “Regardless, such correlations should be studied earlier in life when adolescents initiate with smoking and drinking (i.e. between the age of 14 and 16 years).“ Please see last paragraph of the subsection 4.2

Reference section

At reference 11 (line 434), the doi is missing.

and

At reference 14 (line 442), the doi is missing.

and

At reference 25 (line 472), the doi is missing.

and

At reference 30 (line 488), the year 2003 has been entered twice.

and

At reference 33 (line 498), the doi is missing.

and

At reference 35 (line 503), the doi is missing.

and

At reference 40 (line 515), revise the citation of this reference.

and

I recommend the following website to find the articles' doi: https://search.crossref.org/references

RESPONSE: Thank you. All references are checked and DOI number are added (if applicable).

Staying at your disposal

Authors

Reviewer 2 Report

Overall, the study is interesting, and I think the authors have done a good job. However, I think there are some shortcomings that should be resolved before being accepted for publication.

Abstract:

The abstract begins: “Concurrent smoking and harmful drinking (CSHD) in adolescence is an important public health- and social-problem. This study aimed to prospectively evaluate the influence of sports participation on the prevalence of and initiation into CSHD of adolescents from Croatia”. However, it is later indicated that: “A higher risk of CSHD at baseline was seen among those adolescents who were involved in sports for longer (OR = 1.66, 95%CI = 1.16-2.38). The results are discussed in the context of the fact that the study included adolescents at the age of rigid sports selection (the transition from youth to professional-level sports).” So, the specific goal is unclear. Because at first they talk about public health, but later about sports transition. I kindly consider that it should be harmonized for a better understanding.

In addition, I believe that the objective of the investigation should be more specific. Throughout the study, aspects that are not previously clear when the abstract is read are analyzed. I propose to consider including aspects such as gender, the type of sport or the level of training, since it would provide more information about what will later be developed in the study.

Introduction:

I think the introduction is great. I congratulate the authors. However, the authors comment that: "Binomial logistic regression with age and gender as covariates suggested ..." therefore, if age and gender have been covariates, it should have been justified in the introduction how these variables can influence the question studied.

Also, it is observed in the results that they are divided according to: individual sport and team sport. Therefore, possible differences should be argued that will later be reflected in the results.

Likewise, the results show groups based on the level of training or competition. Therefore, this should also be reflected in the introduction.

Materials and Methods:

I suggest that the information contained in the "procedures" section be divided into two sections: Instruments (or measures) and procedure.

Discussion:

The discussion should begin by remembering the objective of the investigation.

In general, the discussion should follow a story line that has previously been justified in the introduction. That is, aspects such as gender, type of sport, level of training, etc. are discussed. This should be made clearer in the introduction and in the purpose of the investigation. I kindly suggest you review this throughout the paper.

Author Response

REVIEWER 2

Overall, the study is interesting, and I think the authors have done a good job. However, I think there are some shortcomings that should be resolved before being accepted for publication.

RESPONSE: Thank you for your support. We have followed your comments and amended the manuscript accordingly. Please see responses in the following text.

Abstract:

The abstract begins: “Concurrent smoking and harmful drinking (CSHD) in adolescence is an important public health- and social-problem. This study aimed to prospectively evaluate the influence of sports participation on the prevalence of and initiation into CSHD of adolescents from Croatia”. However, it is later indicated that: “A higher risk of CSHD at baseline was seen among those adolescents who were involved in sports for longer (OR = 1.66, 95%CI = 1.16-2.38). The results are discussed in the context of the fact that the study included adolescents at the age of rigid sports selection (the transition from youth to professional-level sports).” So, the specific goal is unclear. Because at first they talk about public health, but later about sports transition. I kindly consider that it should be harmonized for a better understanding.

and

In addition, I believe that the objective of the investigation should be more specific. Throughout the study, aspects that are not previously clear when the abstract is read are analyzed. I propose to consider including aspects such as gender, the type of sport or the level of training, since it would provide more information about what will later be developed in the study.

RESPONSE: Thank you for accurate explanation and suggestion. Indeed, in the original version of the Abstract the objectives were not clearly identified but we must mention that this was mostly because of the word limit (200 words max). In this version of the manuscript abstract is systematicall rewritten, and we believe that it is more focused and clear. We tried to identify all neccesary aspects of the study, and abstract now reads: “Concurrent smoking and harmful drinking (CSHD) in adolescence is an important public health- and social-problem, while participation in sports is considered as being protective against CSHD This study aimed to prospectively evaluate the influence of various facets of sports participation on the prevalence of and initiation into CSHD of adolescents.. Participants were adolescents from southern Croatia (n = 711, 43.6% females, 16 years of age at study baseline), who were tested at baseline and at follow-up (two years later): Variables included gender, age, sports factors (participation in individual- and team-sports, sport-experience, -competitive success, intensity of involvement in sports), and CSHD. The CSHD prevalence did not increase significantly over the course of the study (from 5.6% to 7.5%, p > 0.05). Binomial logistic regression with age and gender as covariates suggested that team sports participation correlated to CSHD prevalence at baseline, and follow-up, with higher risk for CSHD among those adolescents who quit team sports (OR = 9.18 and 2.68, 95%CI = 2.04-22.26 and 1.05-6.83 for baseline and follow-up, respectively), and those never involved in team-sports (OR = 9.00 and 3.70, 95%CI = 2.07-39.16 and 1.57-8.72 for baseline and follow-up, respectively). A higher risk of CSHD at baseline was seen among those adolescents who were involved in sports for longer (OR = 1.66, 95%CI = 1.16-2.38). The results are discussed in the context of the fact that the study included adolescents at the age of rigid sports selection (the transition from youth to professional-level sports). Since the majority of participants began CSHD at an earlier age, further studies in subjects of a younger age range are warranted.”

Introduction:

I think the introduction is great. I congratulate the authors. However, the authors comment that: "Binomial logistic regression with age and gender as covariates suggested ..." therefore, if age and gender have been covariates, it should have been justified in the introduction how these variables can influence the question studied.

RESPONSE: Thank you for your support. Following your suggestion in this version of the manuscript we briefly explained the necessity of controlling age and gender as covariates in logistic regression. This is done in Methods (Statistics). Text reads: “The logistic regression calculations were adjusted for gender and age because previous studies identified significant influence of age and gender on substance misuse in adolescence [6,9]

Also, it is observed in the results that they are divided according to: individual sport and team sport. Therefore, possible differences should be argued that will later be reflected in the results.

RESPONSE: The possible differential effects of individual- and team-sports on substance misuse in adolescence is now explained in the Introduction section. Text reads: This is particularly evident with regard to participation in individual- (i.e. athletics, swimming) and team-sports (i.e. football/soccer, basketball, handball), and studies repeatedly found differential effects of participation in these two type of sports on substance misuse in adolescence [9,31]. In brief, participation in team-sports was repeatedly associated with higher likelihood of alcohol drinking (because of the post-sport social gatherings) [32]. On the other hand, athletes involved in some individual sports were often in risk for smoking (i.e. in some aesthetic sports are concerned on body weight, because smoking increases basal metabolism) [24,25]. Please see 5h paragraph of the Introduction.

Likewise, the results show groups based on the level of training or competition. Therefore, this should also be reflected in the introduction.

RESPONSE: Thank you. This issue is also explained and text reads: “Also, other facets of sports participation that are found as being specifically related to substance misuse in puberty include experience, level of sport competition (i.e. sport achievement), and intensity of training (i.e. commitment to sport), altogether highlighting the necessity of more profound studying of the problem [27,31]. “

Materials and Methods:

I suggest that the information contained in the "procedures" section be divided into two sections: Instruments (or measures) and procedure.

RESPONSE: Thank you for your suggestion. In this version of the manuscript the original subsection “Procedures” is now divided into “Procedures” and “Instruments and variables”.  IN the first part we explained the testing sequences and procedure of testing. In the second part Instruments and Variables are specifically explained.

Discussion:

The discussion should begin by remembering the objective of the investigation.

RESPONSE: The presentation of main study aims is now added at the beginning of the Discussion. Text reads: “This study aimed to evaluate prevalence of CSHD, and sport factors associated with CSHD in adolescents from Croatia.“

In general, the discussion should follow a story line that has previously been justified in the introduction. That is, aspects such as gender, type of sport, level of training, etc. are discussed. This should be made clearer in the introduction and in the purpose of the investigation. I kindly suggest you review this throughout the paper.

RESPONSE: We must agree that some aspects of the study were not sufficiently covered in the Introduction. We hope that the changes we made (please see previous responses for details) improved this aspect of the manuscript and that the Discussion is now better connected to statements from the Introduction.

Staying at your disposal

Authors

Round 2

Reviewer 2 Report

The authors have adequately responded to the suggestions made.